# 3D printing of ultra-high viscosity resin by a linear scan-based vat photopolymerization system

Zixiang Weng ®[1,2] ✉, Xianmei Huang[1], Shuqiang Peng[1,3], Longhui Zheng[1,4] & Lixin Wu[1,2] ✉

The current printing mechanism of the bottom-up vat photopolymerization 3D printing technique places a high demand on the fluidity of the UV-curable resin. Viscous high-performance acrylate oligomers are compounded with reactive diluents accordingly to prepare 3D printable UV-curable resins (up to 5000 cps of viscosity), yet original mechanical properties of the oligomers are sacrificed. In this work, an elaborated designed linear scan-based vat photopolymerization system is developed, allowing the adoption of printable UV-curable resins with high viscosity (> 600,000 cps). Briefly, this is realized by the employment of four rollers to create an isolated printing area on the resin tank, which enables the simultaneous curing of the resin and the detachment of cured part from the resin tank. To verify the applicability of this strategy, oligomer dominated UV-curable resin with great mechanical properties, but high viscosity is prepared and applied to the developed system. It is inspiring to find that high stress and strain elastomers and toughened materials could be facilely obtained. This developed vat photopolymerization system is expected to unblock the bottleneck of 3D printed material properties, and to build a better platform for researchers to prepare various materials with diversiform properties developed with 3D printing.

The development of automotive, aerospace, bioengineering, soft robotics, and electronics is inseparable from the synthesis and processing of high-performance polymers (HPP) materials[1]. Structural diversification requires technological advances for the controllable preparation of HPP with heterostructures. Polymer-based additive manufacturing, also known as 3D printing technique, has received extensive attention and rapid development due to its high flexibility in 3D architecture building. Among various 3D printing techniques, vat photopolymerization (VPP) is one of the most promising technologies[2] in view of its advantages of high-efficiency, large build envelope, economic, and versatile material options[3–7]. Besides, the VPP technique also provides the highest fabrication accuracy to prepare parts with lower weight and higher strength[8].

VPP, a 3D printing technology that use UV light to cure UV-curable resins, can be roughly classified as top-down fashion (represented by stereolithography, SLA) and bottom-up fashion (represented by digital light processing, DLP) according to the platform movement direction[9]. Different from the conventional SLA technology that demands large quantity of the resin for starting material, DLP 3D printing technology occupied smaller footprint, and is more material-saving and economical, which makes it widely used in academy research and high-precision printing. Moreover, compared to SLA which employs a single

[1]CAS Key Laboratory of Design and Assembly of Functional Nanostructures, Fujian Key Laboratory of Nanomaterials, Fujian Institute of Research on the Structure of Matter, Chinese Academy of Sciences, Fuzhou 350002, PR China. [2]Fujian Science & Technology Innovation Laboratory for Optoelectronic Information of China, Fuzhou, Fujian 350108, PR China. [3]Key Laboratory of Polymer Materials and Products, College of Materials Science and Engineering, Fujian University of Technology, Fuzhou 350118, PR China. [4]Present address: Fujian Science & Technology Innovation Laboratory for Optoelectronic Information of China, Fuzhou, Fujian 350108, PR China. ✉e-mail: wzx@fjirsm.ac.cn; lxwu@fjirsm.ac.cn

laser point, DLP 3D printing technology achieves a resolution of about 1 μm regardless of the layer's lateral area and complexity[10]. Yet, there are still some challenges for VPP technology waiting to be addressed. For example, both SLA and DLP 3D printing technologies are facing the limitation of efficiency. Also, the large exposure area during printing needs to be compromised by changing the orientation of the model. Besides, printed part behaves mechanically anisotropy in vertical and horizontal directions[11], which is affected by the degree of polymerization and post-print curing process[12].

Significant efforts have been devoted to the upgrade of the VPP technique. Carbon Inc. which invented "CLIP" technology[13] and EnvisionTEC[14] incorporated an oxygen-permeable window above the ultraviolet image projection plane to realize a non-stop manufacturing process by inhibiting polymerization of the UV-curable resin in the "dead zone". Also, Walker et al. [15]. created a slip boundary between the resin and the window using fluorinated oils to realize large volume continuous printing. Above approaches eliminate the waiting time for ascending and descending of building platform. Although these continuous elevation printing features greatly improved the preparation efficiency, a lower viscosity (< 500 cps) for starting material is required to ensure the liquid resin flows into the small gap between the cured layer and the bottom of the resin tray[16]. Therefore, diluents[17] or reactive monomers[18] are commonly introduced in the viscous oligomer system to balance the demanded viscosity and desired properties. However, diluents and monomer are volatile and irritating, and the introduction of a high proportion of which would sacrifice the original mechanical properties of the oligomer[19] and cause the volumetric shrinkage of the printed parts[20].

Acrylate oligomers hold great promise in VPP since they can exhibit desired properties by molecular structure design, yet the relatively high viscosity of which limits their printability in currently used VPP technologies[21–23]. Specifically, the UV-curable acrylate resins for bottom-up fashion VPP are categorized as monomer, epoxy acrylate oligomer and polyurethane acrylate oligomer. Among which, the chemical structure of polyurethane acrylate is rich in the structure selection. By adjusting the ratio and phase separation of soft and hard segments, the preparation of materials from flexible to toughened can be realized[24]. However, the satisfied mechanical properties of the polyurethane acrylate oligomers originate from their high molecular weight, which relates to a high viscosity[25]. For example, the oligomers with high molecular weight facilitate the formation of highly entangled polymer chain, contributing to the preparation of high-elastomeric material with improved strain and stress. Although the elastomers prepared by VPP have been reported, most of the work required the addition of high content (40% w/w) of reactive diluents[19] and deteriorated the high resilience performance of the original polyurethane acrylate oligomer. For toughened material prepared by VPP, the soft segment in the polyurethane acrylate oligomer usually employs a crystalline moiety such as polycaprolactone diol. Although the crystallinity of the soft segment is conducive to improving the material strength and toughness of the material, the resin is usually crystallized at room temperature due to the crystallinity[26], which still needs to be heavily diluted before applying in VPP.

Consequently, the key to improving the mechanical properties of the printed part is to break the low-viscosity limitation of processable resin. The key technical difficulty of 3D printing of high-viscosity resins lies in the leveling of the resin and deformation of the cured parts from the film. Conventional 3D printing techniques adopt a sweeper to recoat the resin as seen in the SLA system[27], yet the excessive shear force affects the original position of cured part, leading to a fabrication failure. It has been uncovered that the viscosity of the resin must still remain below 5000 cPs for an SLA 3D printer[28]. For DLP 3D printer, the printing of high-viscosity resin is also realized by a recoater (3D printers from Lithoz Inc. for instance[29]) or a film tape[30]. Nevertheless, both methods did not address the huge detachment force when printing a layer with a large horizontal exposure area. For both SLA and DLP 3D printer, thermal treatment of the resin may reduce the operational viscosity for use, but the poor thermal conductivity of the UV-curable resin limits the heating effects. For epoxy-dominated resin, heating also promotes UV-curable resin experiencing a "dark reaction" and accelerates the crosslinking reaction, thereby shortening the service life of the UV curable resin[31].

The goal of this work is to assemble a 3D printer that can address following issues: printing of high-viscosity UV-curable resins also showing a compatibility with the commercial low-viscosity resins at room temperature; printing of bulk parts with a large exposure in horizontal plane; printing objects with excellent performance by using commercialized oligomers. Toward these targets, an elaborated designed linear scan-based vat photopolymerization (referred as LSVP from here on) technique is proposed. Briefly, LSVP system adopts a homemade linear scan laser modulus supplemented by low surface energy release film (made by fluorinated ethylene propylene, FEP) and rollers, which can greatly broaden the printing windows by increasing the viscosity tolerance of VPP to materials while ensuring printing accuracy. This designed LSVP system aims to strike a balance between effectiveness and efficiency, not only to process high viscosity UV-curable resin, but also ensuring that the efficiency is maintained. A brief illustration is shown in Fig. 1. High viscosity UV-curable resins were also prepared to verify the processability of the LSVP system, and corresponding physicochemical properties characterization of printed samples were conducted to demonstrate the merits of the high viscosity UV-curable resins.

## Results and Discussion
### Prototype assembly
At this stage, limited by the high focal length for DLP 3D printer projection equipment, it is difficult for surface exposure-based 3D printers to cast large-scale exposure areas, hence the preparation volume in horizontal plane of products is limited compared to SLA 3D printer. Considering the footprint waste and high price for DLP chips, the goal of the laser module design is to assemble a set of linear laser scanning system which is small, economic, and stable. The linear scan strategy has been previous reported by Mao et al. used in a "LISA" system[32], which can effectively eliminate the recoating and separation time. Also, such strategy was also employed in the vector 3SP 3D printer developed by EnvisionTEC, which delivers a higher resolution and larger build envelope[33]. However, the printing of high viscosity UV-curable resin has not been reported. In this work, a smaller size laser module which emits a 405 nm wavelength light beam was adopted in the LSVP system, which is shown in Fig. 1a, and the physical photo is shown in Fig. S1. It should be noted that to eliminate the uneven energy distribution errors brought by the start and stop time of conventional two-axis gyro-mirror system, in this work, a reflecting prism was adopted which is simpler and sufficient to reflect demanded laser. The laser shoots at a high-speed rotating reflecting prism whose rotation speed is a constant of 16,000 rpm. As the refraction angle changes, the point light source forms a scanning line beneath the resin tank. Through the movement of the laser module along the x-axis (Fig. 1b), this scanning line draws a specific two-dimensional image. Because the power of the laser module is fixed, the exposure energy can be varied by adjusting the repeat times for one scanning line. This prototype design combines the solution of reflecting prism and laser module together to realize large-scale printing. Corresponding video illustrating the working mechanism of the laser module is shown in Supplementary Movie 1.

The framework of the resin tank in this work is 3D printed by an SLA 3D printer (Protofab, Xiamen, China). A fluorinated ethylene propylene (FEP) film was folded to a box shape and fixed by screws in the resin tank. The whole resin tank was fixed by four magnets attached beneath the edge of the resin tank. In this work, the available printing size is 120 mm (x direction) × 108 mm (y direction) × 160 mm

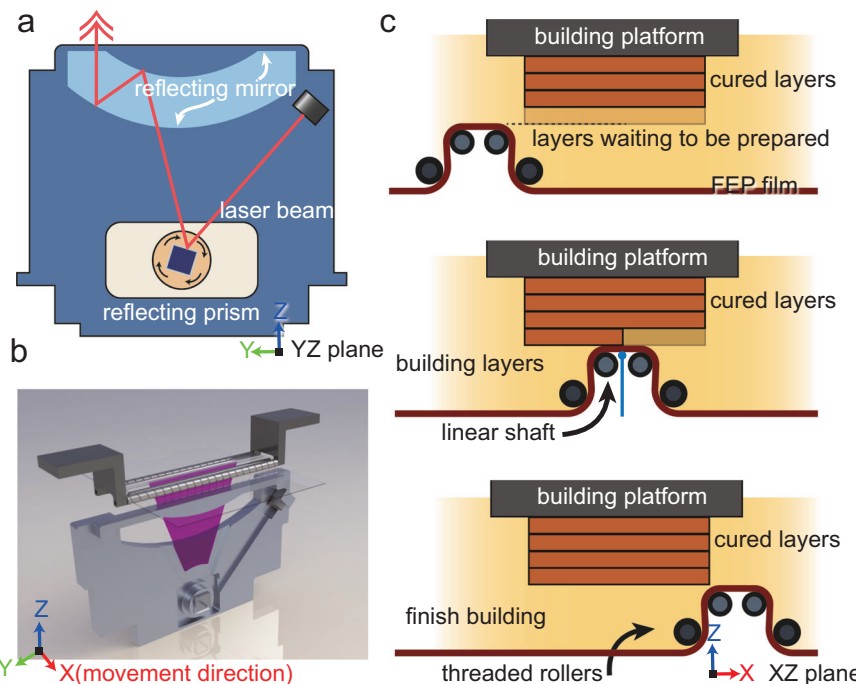

**Fig. 1 | Illustration of LSVP. a** Illustration of the laser device in the *yz*-plane. The centered blue square represents the reflecting prism. **b** Illustration of the entire LSVP structure, where the purple sector represents the scanning line. The laser device moves along the *x*-direction. **c** Illustration of the LSVP roller system in the *xz*-plane. The two smaller grey circles represent the linear shaft, while two larger grey circles represent the threaded rollers. The brown curve in the bottom represents the transparent FEP film. The blue vertical line represents the laser beam.

(*z* direction) for the prototype machine. The engineering drawing including the sizes of the rollers are shown in Fig. S2. The video of the resin tank setup is illustrated in Supplementary Movie 2.

The test samples provided in this work are directly printed by LSVP system instead of casting (printed samples are shown in Fig. S3). Detailed information are illustrated in Methods section. To eliminate the errors resulting from the temperatures, all printing experiments were performed at 20 °C. For the low-viscosity resin, the level surface of the resin should be higher than the linear rollers, otherwise insufficient resin remains on the top of the lifted dome. Nevertheless, for the high viscosity resin, the level surface was expected to be lower than the linear shafts. Due to the roller movement, the flow of the high-viscosity resin will apply a considerable force on the cured part, resulting in an offset in the z axis direction. To eliminate this error, only 100 g of high viscosity resin were adopted in this work. All printed files are properly oriented and can be downloaded in Supplementary Files.

## Analysis of LSVP system

A fluorinated ethylene propylene (FEP) film, with a low surface energy and favorable light transmittance, was folded into a box shape and fixed at the bottom of the resin tank to hold the liquid resin (Fig. 1c). Different from the conventional bottom-up 3D printing technique, in the LSVP system, a large deformation of FEP film is realized by the adoption of four rollers comprising two linear shafts beneath the film and two threaded shafts above the film. When printing high-viscosity resin, the advancing threads rollers can evenly spread the high-viscosity resin on the film of the resin tank, then perform laser curing, and a linear detachment will be realized between the cured parts and the film at the bottom of the resin tank. The three actions of "spreading", "curing", and "detachment" are completed simultaneously. Compared to the conventional DLP 3D printer, the special way of "detachment" of LSVP delivers a higher print speed and longer service life of the films. In a LSVP system, the film with a large deformation applies a smaller linear detachment force to peel off the printed part from the film. A classic bottom-up 3D printing technique employs a straight pull detachment

fashion to realize the detachment for cured part and the film. This surface detachment approach makes the orientation and shell extraction of printed parts critical. Not only the considerable volumetric shrinkage of acrylate resin, but the enormous detachment force will shorten the service life of the film. For the case of printing a solid cylinder in a bottom-up 3D printer, a "adhesion of plane surface" model can be introduced to describe the detachment behavior[34]. The pull force is related to the contact area in *xy*-plane according to Eq. (1)

$$P = \sqrt{\frac{8\pi}{(1-v^2)}E\gamma a^3} \tag{1}$$

where $P$ is pulling force, $v$ is the Poisson's ratio, $E$ is the Young's modulus of FEP film, $\gamma$ is the interfacial surface energy defined as the energy required to separate unit area of the contacting surfaces, $a$ is the radius of the cylinder. From Eq. (1), it showed that the pull force is proportional to the 1.5th power of the radius. Above mentioned equation indicates that when printing a solid structured object, it is necessary to tilt the orientation of the object to reduce the exposure area on the *xy*-plane in a single layer preparation.

However, the behavior of the linear detachment in an LSVP system can be described by the "peeling of thin films" model, where the cured part and deformed FEP film are treated as a rigid substrate and an elastic film, respectively. This behavior can be described by the Eq. (2):

$$P = \frac{b\gamma}{1 - \sin\theta} \tag{2}$$

where $P$ is the linear detachment force, $b$ is the length of the contact area on the y-axis, $\theta$ is the angle between the z-axis and pulling direction. Equation (2) reveals that the pulling force is only proportional to the first power of the length, representing that the linear detachment force has no relationship to the contact area. Meanwhile, in a linear detachment way, the $\theta$ is ca. 50°, which makes the linear detachment force reach to a minimum value resulting in a longer service-life of the

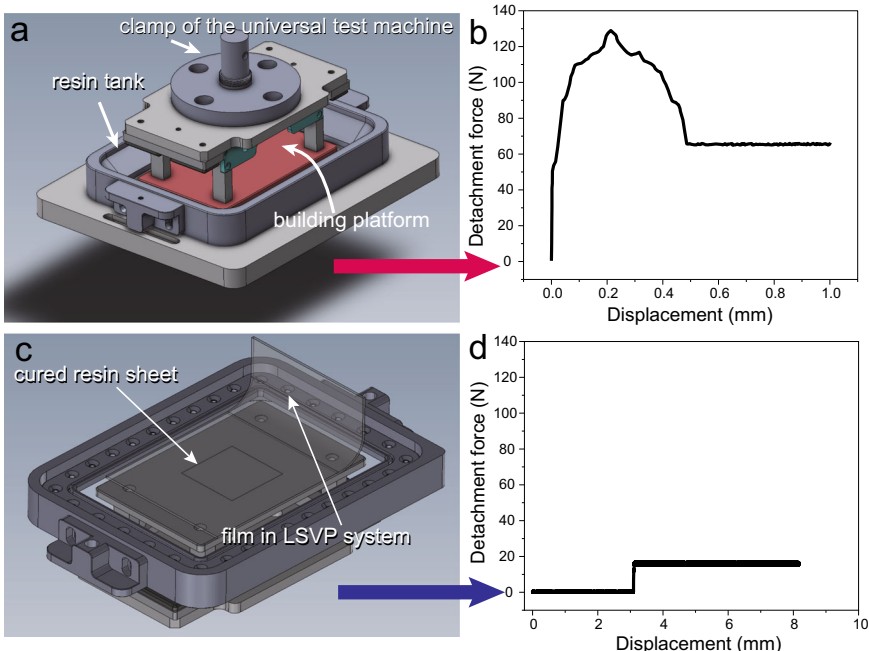

**Fig. 2 | Detachment force validation and comparison between conventional DLP 3D printer and LSVP system. a** Detachment force validation setup for conventional DLP 3D printer. **b** Detachment force variation during direct pulling-up. **c** Detachment force validation setup for LSVP system. **d** Detachment force variation during peeling-off in LSVP system. Source data are provided as a Source Data file.

film. When printing an elastomer, the linear detachment method also overcomes the flaw of cured part deformation caused by the straight up and down detachment method (Fig. S4).

To validate the advantages of the peeling-off fashion in LSVP, a simulated scenario for detachment force test was carried out. Here, the modified clamps of the universal material testing machine were employed to compare the differences in detachment force between the straight pull detachment fashion in a conventional DLP 3D printer and the linear peeling-off detachment fashion in the LSVP 3D printer. As shown in Fig. 2a, a fixed area (50 mm × 50 mm) of high-viscosity UV-curable resin was poured into the resin tank and cured by a DLP projector and put the whole set of fixtures on the universal material testing machine for stretching test to simulate the detachment process of conventional DLP 3D printing. For the LSVP system, the FEP membrane at one end was directly clamped on the fixture of the universal testing machine for tensile testing, as shown in Fig. 2c. Both physical photos can be seen in Fig. S5. The results show that for conventional detachment fashion, the pull force gradually increases as the experiment running. When the FEP film and the cured resin system are completely separated, the pull force reached the maximum value (130 N) and dropped back to a constant (Fig. 2b). In contrast, the unique linear peeling-off method adopted by LSVP shows a more gentle and stable force (Fig. 2d) while the peeling value is stable at 18 N. From the integral area of the two curves, it can be seen that compared with the DLP 3D printing technology, LSVP system did less work when printing a large-area *xy* plane, which is beneficial to prolong the service-life of the release film.

CLIP technology can effectively eliminate the detachment force, yet the self-leveling time is closely related to the viscosity of the resin according to exposure area. To make a comparison between CLIP and LSVP technology, a solid square model for example was adopted to evaluate the printing speed. According to Eq. 3: (See the Supplementary Information for derivation details),

$$t = \sqrt[3]{\frac{3\mu a^2}{\triangle P \cdot v^2}} \qquad (3)$$

where *t* represents the time for self-leveling, *μ* represents the apparent viscosity of the resin, it can be concluded that the printing of CLIP technology is heavily reliant on the viscosity of the resin, specifically, the higher the viscosity of the resin is, a longer self-leveling time is. Nevertheless, with high viscosity resin or exposure a large area in the *xy*-plane, the elevation speed had to be decreased to fulfill the self-leveling requirement. Whereas for the LSVP system, the printing speed is only related to the movement speed, and the exposure energy is proportional to the number of scans. In this prototype design, the movement of the laser module is 35 mm/s if the number of scans is one. For those resins with low reactivity, increasing the scan number will emit higher energy for curing. By increasing the scan number to *n*, the movement of the laser module v = 35/*n* mm s⁻¹. Corresponding illustration is shown in Fig. S6. Hence the printing time is only related to the length in *x* axis and the reactivity of the UV-curable resin.

In conclusion, LSVP system with the feature of synchronization of fabrication and linear detachment can eliminate the steps of elevating and descending of the resin base. In addition, if the model has a smaller span on the *x*-axis, the efficiency of printing can be further improved. Also, the print speed of LSVP is irrelevant to the exposure of *xy*-plane. Most importantly, the orientation of the printed sample can be freely rotated without the consideration of the *xy*-plane. In the following works, to investigate what the ultimate mechanical properties a UV-curable resin can achieve after breaking the viscosity limit, two types of polyurethane acrylate – elastomer and toughened materials with high weight proportion of oligomer were prepared, also, commercial 3D printing resin with low viscosity was employed to validate the processibility of LSVP system.

## Printing of elastomer with high viscosity resin

While the LSVP system is capable of fabricating large bulk and refined objects, its superiority over other printing technologies stems from its ability to print various types of UV-curable resins with high viscosity. Recently, the interest in additive manufacturing of elastomers is rapidly growing because of the wide range of potential applications[35]. For 3D printed elastomer, although latent curing agents can usually produce an elastomer

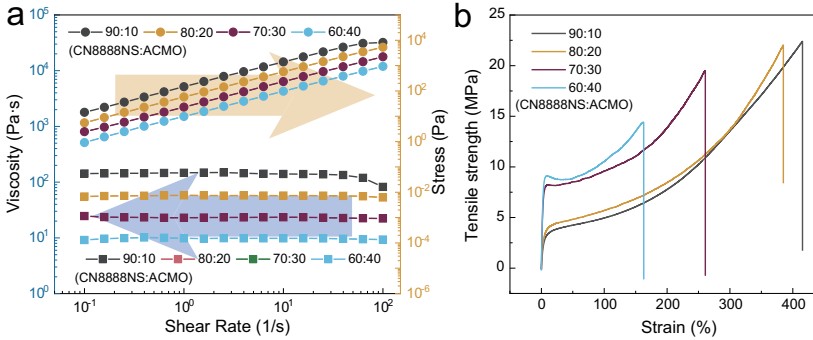

**Fig. 3 | Physical and mechanical properties of oligomer-dominated UV-curable resin. a** The shear rate versus viscosity of mixtures with CN8888NS series samples. Relative proportions of the UV-curable resin are listed in the legend.

**b** Stress-strain curves of printed CN8888NS series samples. Source data are provided as a Source Data file.

with higher elongation at break and higher tensile strength[36], the insufficient thermal stability of as-prepared uncured resins would cause a shortened shelf life. Based on this consideration, this work only investigates single component-based UV-curable resins. The mechanical properties of polyurethane containing isocyanate and polyols have been studied widely in the field of 3D printing, however corresponding reports relating to polyurethane acrylate in the field of 3D printing are rare due to the formation of stronger hydrogen bonds between backbones leading to a higher viscosity[37]. In this work, the LSVP system was used to fabricate a high-viscosity polyurethane acrylate resin to study how the viscosity affects the mechanical properties. Here, CN8888NS from Sartomer was adopted as a high viscosity oligomer, which provides a very high stress and strain (i.e., a stress of 20.46 MPa, a strain of 963% and a viscosity ranged from 25,000 to 30,000 cPs at 60 °C[38]). The introduction of difunctionalized reactive diluents usually brings a higher rigidness for the printed part. Hence, in this work, the physicochemical properties, including mechanical properties, volumetric shrinkage and storage modulus etc. of UV-curable resins with different proportion of 4-acryloylmorpholine (ACMO) and CN8888NS were compared.

Corresponding data of viscosities of different formulations of ACMO and CN8888NS are compiled in Fig. 3a. When the content of ACMO is 10 wt%, the viscosity of the mixture reached 142,786 cps at room temperature. With an increase in ACMO content, the viscosity gradually decreased from 55,113 cps (20 wt% of ACMO) to 9158 cps (40 wt% of ACMO). It also should be noted that the viscosity of the mixture merely varied with the increase of the shear rate. Compared with typical UV curable resins (i.e., high reactive diluent and a low viscosity), the functional group density is significantly lower for the oligomer dominated UV curable resin, resulting in a slower reaction rate (corresponding curing speed curves are shown in Fig. S7). However, from the FTIR spectrum obtained before and after printing a single layer of oligomer dominated UV-curable resin (Fig. S8), the complete disappearance of peaks attributed to the twisting vibration around 810 cm$^{-1}$ proved a high conversion rate of C=C bonds[39]. As shown in Fig. 3b, UV-curable resin comprising high content of CN8888NS exhibited excellent mechanical properties, particularly when the content of ACMO reached 10 wt%. The strain of the specimen is higher than 400%, and the strength is higher than 20 MPa. When compared with thermoplastic polyurethane[40], this elastomer prepared using a high content of oligomer provided a higher strength and a comparable strain. Also, the volumetric shrinkage of CN8888NS series samples (Fig. S9) showed that a higher proportion of oligomer brings a lower volumetric shrinkage. Other data including DMA results, curing depth, critical exposure energy etc. are compiled in Fig. S10 and Fig. S11.

Considering that the increase of the ACMO from 10% to 20% would not cause severe deterioration, the mixture with ACMO weight content of 20% was chosen for further testing. In these tests, complex models including the classic Eiffel Tower model (printing procedure see Supplementary Movie 3), a hexagon solid part with a layer thickness of 100 μm, and a lattice structure with a layer thickness of 25 μm were selected to verify the printability of the high viscosity resin without the limitation of structure and orientation of the part. In Fig. 4a, the printed Eiffel Tower model retained all its detailed structures. Due to the ductility of the used resin, the top portion of the Eiffel Tower can be twisted freely without breaking. Also, a solid hexagon and the elaborated lattice structure shown in Fig. 4b, c, were both successfully printed while the lattice with a favorable resilience property was maintained. Figure 4d represents the scanning electron microscope image of printed lattice structure. The distinct layer shifting indicates that LSVP has the capability to print refined structure with high viscosity resin.

## Printing of toughened materials with high viscous resin

Besides elastomers, 3D printed materials with high toughness are also of interest for industrial applications. The area beneath the stress-strain curves is calculated to evaluate the toughness of a material[41]. In this work, CN8883NS (purchased from Sartomer, China) printed together with ACMO results in components with high toughness. CN8883NS oligomer has a favorable balance between stress and strain[42]. Yet, the relatively high viscosity of CN8883NS makes it unprintable without a reactive diluent. ACMO exhibited a better diluent effect on CN8883NS than on CN8888NS according to the extent of viscosity decrease. As shown in Fig. 5a, The viscosity decreased from 600,000 cPs to 20,000 cPs with an increase in the ACMO content from 20 wt% to 50 wt%. While a mixture of this viscosities still cannot be used in a DLP 3D printer, it can be applied to the LSVP system. From the FTIR spectrums of CN8883NS series samples (Fig. S12), it can also be proved that the conversion of the double bonds is complete.

Figure 5b, c represent the tensile strength and the flexural strength of the 3D printed samples, respectively. To assess the LSVP system with respect to producing samples with high toughness, the results from ABS parts prepared through injection molding are shown in Fig. 5c. The flexural test procedures were recorded in Supplementary Movie 4. By comparison, the flexural samples prepared by CN8883NS together with ACMO can be folded (shown in Fig. 5d), and the maximum strain (75.0 MPa) and flexural modulus (2.17 GPa) are very close to injected ABS (72.0 MPa and 3.33 GPa, respectively). Also, they share a similar area beneath the stress-strain curve, (15190 for the 3D printed sample, and 16032 for the injected ABS sample) suggesting they possess similar toughness. Figure 5e and Supplementary Movie 5 demonstrate how the as-prepared resin can be carved by a knife without brittle fractures as would be seen for other engineering plastics like ABS. Similarly, the volumetric shrinkage, storage modulus and curing depth of CN8883NS series samples were also characterized, which exhibit a similar trend like CN8888NS. Corresponding data are compiled in Figs. S13–S15.

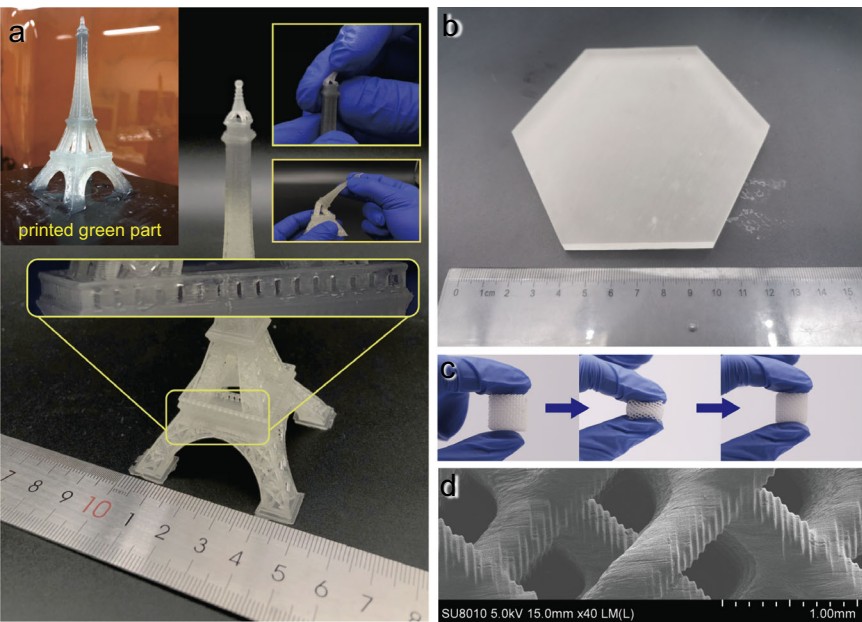

**Fig. 4 | Printed samples by high-viscosity elastomeric UV-curable resin.**
**a** Printed Eiffel Tower model using CN8888NS: ACMO = 80:20. The refined structure including the fence and the top platform of the tower can be printed accurately, and the model can be twisted or bended freely (printing layer thickness is 0.1 mm). **b** Printed solid hexagon. The thickness of the hexagon is 7 mm, and the side length is 60.62 mm. The layer thickness is 0.1 mm. **c** Printed lattice structure exhibited favorable resilience. **d** Scanning electron microscope of printed lattice structure.

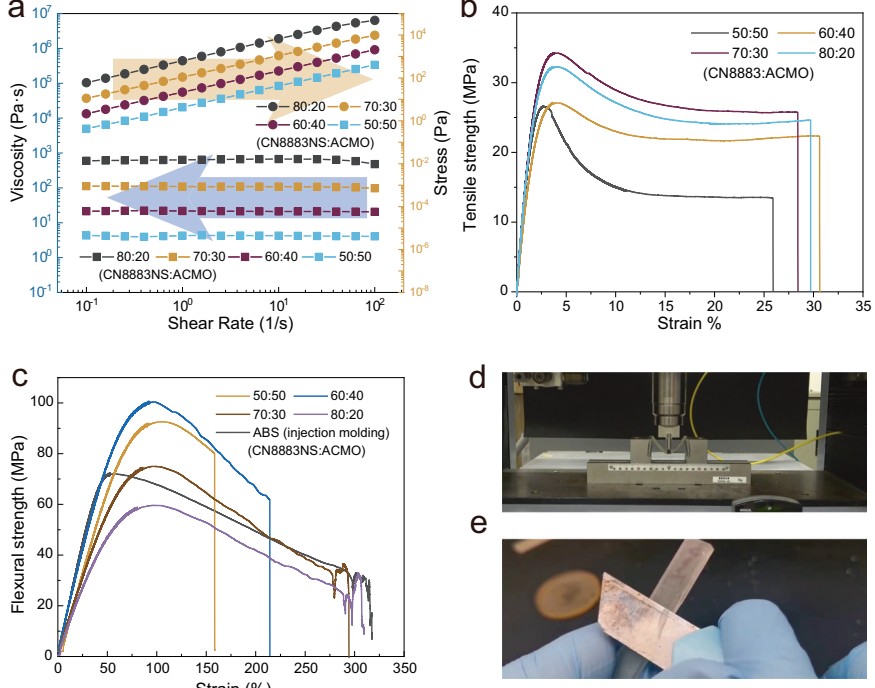

**Fig. 5 | Physical and chemical properties of the high content oligomer mixture.**
**a** The shear rate versus viscosity of mixtures with different proportions. 102 g of each sample were prepared containing 2 g of TPO and different proportions of ACMO and CN8883NS. Relative proportions are listed in the legend. **b** Tikhe tensile stress-strain curves of printed samples. **c** The flexural stress-strain curves of printed samples. **d** Samples of CN8883NS: ACMO = 70:30 bending result. **e** Sample carving test. Source data are provided as a Source Data file.

## Printing of commercial low viscosity resin

To verify whether this proposed fabrication system is capable of printing parts with either high accuracy or large volume, a low viscosity resin (450 cps) was adopted to evaluate the compatibility of the LSVP system. The detailed formulation of the resin is listed in Supplementary Table 1, while the curve of tensile strength versus strain is displayed in Fig. S16. From results showed in Fig. S15, it can be seen that the samples prepared by vertical printing or side printing have high consistency in terms of Young's modulus, elongation at break and tensile strength,

which also shows that both the detachment fashion and the linear scanning method of the LSVP system can be applied to the printing of commercial low viscous UV-curable resins.

Meanwhile, a series of bench-structured models with various lengths of span, precision test models and an Eiffel Tower model were chosen for further printing. From Fig. 6a, it can be observed that using a commercialized DLP 3D printer, the cured part, directly pulled up from the bottom of the resin tank, leads to a deformation of the cured thin layers. In addition, the surface exposure causes the acrylate to shrink considerably, which affects the dimensional accuracy. The carved characters on the surface also became blurred and difficult to recognize. Enlarged pictures of the printed structures are shown in Fig. S17.

In comparison with the conventional DLP 3D printer, LSVP system exhibits a better surface quality along the z-axis owing to the adopted linear detachment fashion. A digital caliper indicated a 0.5 mm error exists between the actual part and the programmed dimension. This stems from the diameter of the laser spot (~0.2 mm), which can be further eliminated by optimizing the scanning line programming. By this scanning line programming, the LSVP system combines the advantages of an SLA 3D printer which is capable of fabricating large-scale parts and the advantages of a DLP printer which is capable of printing refined structures. Besides, the dimensional accuracy can be easily adjusted by changing the scan number to match the critical exposure ($E_c$) of different UV-curable resins. More importantly, thanks to the unique detachment fashion of LSVP, a longer bench span without the supports is feasible. In this case, when the span reaches 25 mm, there is still no obvious delamination phenomenon at the bottom of the unsupported span, which indicates that LSVP has a higher tolerance to the orientation of the model.

As shown in Fig. 6B, the LSVP printed a precision test model with a brush-like structure attached to the platform and a lattice structure with a z axis of 0.025 μm. The edge length of vertical stick increased from 0.05 mm to 0.5 mm in 0.05 mm increments. Insufficient scans is unable to provide minimum exposure energy, leading to a printing failure of the bar whose edge length is smaller than 0.25 mm. By increasing the number of scans to 20, the bars of edge length equal to 0.15 mm were successfully printed. Further increasing the number of scans makes the finest bar printable, but this results in overexposure of

the resin. By properly adjusting the scanning time, a refined mesh with a 0.025 μm thickness can be fabricated. In Fig. 6c, a dense mesh with an area of 118.154 mm × 88.615 mm in xy plane and a height of 119.5 mm in z axis is printed. This model is employed to test a long-term usage for LSVP system. With a conventional DLP 3D printing technique, "cloudy effects" from the detaching film is still a big hinderance for the fabrication of parts with a long distance in z-axis[43]. In a LSVP system, the detachment fashion extends the service life of the film in the bottom of the resin. First, detachment in the LSVP requires a much lower force to remove the part. The feature also lasts longer because the heat generated by polymerization of acrylate dissipates quicker compared to the conventional DLP fabrication fashion. Also, the fabricated Eiffel Tower (Fig. 6d) proved the excellent compatibility for printing a complex geometric structure in the LSVP system. The video recording the whole Eiffel Tower printing procedure is shown in Supplementary Movie 6.

In this work, a facile and time-saving UV-curable 3D printing method called LSVP was proposed and validated. By employing a linear laser scanning modulus, LSVP can spread the UV-curable resin with ultra-high viscosity evenly on the surface of the FEP film, and the UV-curing and detachment of cured parts from the resin tank were performed simultaneously. In the following validation, LSVP system proved to be capable of 3D printing whether high viscosity and low viscosity resin. Corresponding results showed that the LSVP exhibited a high printing accuracy and favorable isotropic mechanical properties. High viscosity UV-curable resins were also prepared to fabricate elastomeric and toughened objects. It is found that without the limitation set by the viscosity, oligomer dominated UV-curable resins exhibit superior mechanical properties. The LSVP also delivers a high printing accuracy for high viscosity UV-curable resin. It is expected that with this developed LSVP system, various oligomers could be applied to prepare 3D printable UV-curable resins without the aid of reactive diluents.

## Methods
### Resin preparation
The photoinitiator, diphenyl(2,4,6-trimethylbenzoyl)phosphine oxide (TPO) was purchased from IGM, China. 4-acryloylmorpholine (ACMO)

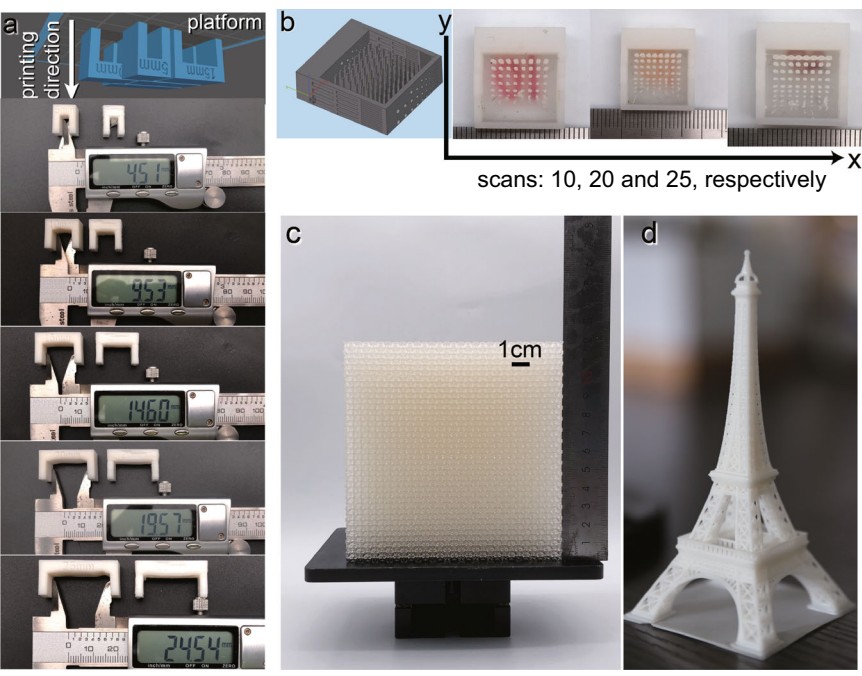

**Fig. 6 | Low-viscosity UV-curable resin printing tests. a** Illustration of the orientation of bench structure and printed bench structure by LSVP system (left) and a DLP system (right). **b** Dimension accuracy test. **c** Fabrication of large lattice object (left) and **d** printed Eiffel Tower model.

was purchased from KJ Chemical, Japan. Oligomer CN8888NS and CN8883NS were purchased from Sartomer (Guangzhou), China. Polyurethane acrylate (NeoRad™ U-25-20D) and bisphenol A epoxy acrylate (CN104NS) were purchased from DSM (Taiwan) and Sartomer (Guangzhou), China, respectively. Other monomers including ethoxylated trimethylolpropane triacrylate (EO3TMPTA) and dipropylene glycol diacrylate (DPGDA) were purchased from Sartomer (Guangzhou), China. Additives including white paste and fumed nanosilica were purchased from Clariant (China) and Evonik (China), respectively. For the preparation of the homogeneous high viscosity resin, TPO was first dissolved in ACMO by stirring for 2 h in a dark room, then mixed with high viscosity oligomer directly using a planetary centrifugal mixer (ZYMC 200, Suzhou ZYE Precision Technology Co., Ltd., China) assisted by a vacuum pump. The rotation procedure is as follows: 450 g (2500 rpm) for 5 min and 140 g (1400 rpm) for 45 seconds. For the preparation of the low-viscosity resin was by stirring mixing resins in proportion together for 0.5 h at a dark room without heating. Corresponding formulation can be found in Table S1 in Supplementary Information.

### Viscosity and photorheology test

A rheometer (Discovery Hybrid Rheometer-2, DHR-2, TA, US) was used to study rheological behaviors of UV-curable resin. The environmental temperature was 25 °C. In this work, a steel made parallel-plate configuration was employed. The diameter of the geometry and the gap between the two geometries were 40 and 0.5 mm, respectively. The range of the steady state shear rate was set from 0.1 to 100 s$^{-1}$. The photorheology behavior was also investigated with DHR-2 attached with a UV LED accessory. The irradiation power of the UV LED light was 80 mW/cm$^2$. The gap between the two geometries was 0.1 mm. The upper geometry was made of aluminum, and the lower geometry was made of transparent PMMA. The experiment lasted a total of 60 s, and the UV LED light was turned on at 20 s.

### Sample preparation and mechanical properties test

The user interface (UI) was programmed by Python. All test samples were designed according to ISO standards (ISO 527/2-5 A for tensile strength test and ISO 178 for flexural strength). For those test samples, samples were directly printed on the platform. For other models, saving designed models as STL file format were oriented in Chitubox basic version to generate necessary supports. An open-source software "nanoslicer" was adopted to slice the preformed 3D model files to PNG files. In this designed system, since the PNG files is unrecognized by the machine, sliced PNG files were converted to.bmp files. To simplify the operation steps, a BAT file was written, and a preformed file only needs to be dragged into it to realize the slicing. The code for the BAT file is listed in the Supplementary Files. The characterizations of the tensile and flexural properties of each printed sample included tensile were carried out with a universal material test machine (AGX-100 plus, Shimadzu, Japan). Specifically, the tensile and flexural behaviors were measured according to ISO 527 and ISO 178 standards, respectively. The environmental temperature was set to 20 °C. A non-contact extension meter was adopted to record the strain during the tensile test.

After printing, the samples fabricated from the low viscosity resin, isopropanol was sufficient to wash away the remaining uncured resin. Regarding the samples fabricated from high viscosity resin, acetone was assisted to wash away the remaining resin in a professional 3D printing green parts washing device (Wash and Cure Plus, Anycubic, China) for 10 s. After cleaning, the samples were further UV-post cured for another 2 min for both sides (Intelli-ray 400, Uvitron, US, 50% of the maximum power).

### Data availability
Source data are provided with this paper.

### Code availability
A BAT code that performs slicing model file to PNG files is provided in the Supplementary Information.

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

## Acknowledgements

The authors want to thank Mr. Jiaxing Shen who assisted the team for designing the UI and providing the support of software development. Also, the authors want to thank Dr. Jia Lin who assisted in revising the manuscript. This work was financially supported by the National Natural Science Foundation of China (Grant No.: 52102148) (Program Manager: L.Z.), The Major Science and Technology Project of Fujian Province (Grant No.: 2021HZ027003) (Program Manager: L.W.), The STS Project of Fujian-CAS (Grant No.: 2022T3071) (Program Manager: Z.W.), and The International Partnership Program of Chinese Academy of Sciences (Grant No.: 121835KYSB20210025) (Program Manager: L.W.).

## Author contributions

Z.W. contributed the conceptualization of 3D printing of high viscosity resin, methodology and original draft preparation. X.H. contributed the preparation of viscous resin, 3D printing and mechanical properties characterization. S.P. and L.Z. contributed the data curation and validation. L.W. contributed to the project administration, funding acquisition, manuscript review and editing. All authors contributed to data interpretation, analysis and drafting of the manuscripts.

## Competing interests

The authors declare the following financial interests/personal relationships which may be considered as potential competing interests: L.Z. and L.W. have patent pending to Fujian Institute of Research on the Structure of Matter, Chinese Academy of Sciences. The remaining authors declare no competing interests.
