## [Peer Review File · Nature Communications]

3D printing of ultra-high viscosity resin by a linear scan-based vat photopolymerization systemEditorial Note: Parts of this Peer Review File have been redacted as indicated to remove third-party material where no permission to publish could be obtained, and to maintain the confidentiality of unpublished data.

REVIEWER COMMENTS

Reviewer #1 (Remarks to the Author):

The paper presents a line scan-based vat photopolymerization (LSVP), which can accomplish "spreading", "curing", and "detachment" simultaneously. This idea has been demonstrated for the same purpose and presented before in a paper, "Linear Immersed Sweeping Accumulation" in the Journal of Manufacturing Processes in 2016. The authors need to compare their approach with the one presented in the JMP paper. The scanning method used is also similar to EnvisionTec's vector 3SP technology. The authors need to articulate the novelty of the paper and how it is different from the previous work as well as why their approach is better.

Also, some more details of the method need to be presented.

(1) The dimensions in Fig. 1C need to be given. What are the sizes of the rollers, and how much are the rollers raised? This will give me an idea of how much the FEP film is deformed.

(2) The tightening of the FEP film from all four sides is required to ensure no resin leaking. Under the film's tension, how much force is needed to deform the film in both X and Y directions is unclear. How to handle the contact points in the Y direction, which seems to have a significant discontinuity?

(3) How such bending forces and continuous scratching/movement will affect the FEP film's life?

(4) The surface portion between the curing line and the left linear shaft in Fig 1C will attach to the cured layer. What are the dimensions? How big is the dragging force in the X direction?

(5) The detaching force measurement in Fig. 3C differs from the setup shown in Fig. 1.

Reviewer #2 (Remarks to the Author):

High-viscosity resin 3D printing is of interest but challenging. This paper develops a VPP method for viscous resin printing by employing a line-shape laser system and four rollers to create an isolated print area that eases the detachment of part from substrate. The claim for being able to print resin with a viscosity of > 600,000 cps is impressive compared to traditional limit of 5,000 cps. However, the introduction section is poorly written without accurate introduction about the fundamentals, without a comprehensive review and comparison to current state of work, and without a coherent flow of information and motivation. Further, the results and discussion section miss details on some key mechanisms.

Please check the comments as follows.

1. Lines 44-51 introduce the DLP's advantage and limitation. Some descriptions or comparisons to SLA are inaccurate. E.g., the paper says DLP has limited throughput, but it is known that DLP is faster than SLA. The authors should also note that not only DLP but also SLA require repeated movement of the build platform. This paragraph needs to be modified to present a more accurate and fair comparison of DLP with other VPP methods and also a comparison of VPP methods with other AM methods.

Btw, the literature review doesn't elaborate on the importance or the need for printing high-viscosity materials. E.g., what are the potential applications given an ability to print viscous materials? The authors should convey the motivation of printing viscous resin better.

2. Line 52: the authors might mention Carbon's technique of CLIP as well, which is well known for its breakthrough in VPP.

3. Please check and fix all the grammar errors.

E.g., Line 64: "odor" is mistakenly used as an adjective.

Line 68: In "yet the extremely high viscosity of which limit their printability in currently used 3D

printer", should "limit" be limits and "printer" be printers?

Lines 69-71: it is hard to understand this sentence: "By eliminating the oxygen inhibition effects, bottom-up fashion provides higher selectivity for the materials, including acrylate monomer, epoxy acrylate oligomer and polyurethane acrylate oligomer and so on."

Line 78: "the DLP printing elastomer"

More other errors throughout the manuscript. The authors should thoroughly proof it.

4. Line 102 (also Line 52) about the paper's goal: What is the specific definition or metric of "throughput" in the authors' goal here? Print speed? Print quality (if so, name the metrics)? If the work aims to increase the print speed for high-viscosity resin, what is the print speed for viscous resins reported in current literature? How does this LSVP speed compare to current speed? Or if throughput means other properties, provide a clear comparison.

5. Addressing the following issues is crucial to accomplishing the LSVP.

Is the light intensity profile uniform as the laser line moves across the build platform? Any projection gap between the scanning laser lines? The authors should discuss the exposure profile in details.

Also, the synchronization between the roller and laser projection needs to be elaborated. How to determine an optimal laser line exposure time and scan speed?

Will the rolling laser module jitter due to the flow resistance from the very viscous resin? Any vibration or non-steady movement of laser module during LSVP? If so, motion blur could be induced to the projected laser line and cause loss of print resolution. Besides, the possible jittering of laser scans may affect the print surface roughness and geometrical features.

6. Some content in Section 2 seems to fit in Section 4 instead.

7. What is the print speed and resolution of LSVP?

8. Any recommendation for future work to improve the LSVP?

Reviewer #3 (Remarks to the Author):

Wu et al have presented an interesting academic paper describing a new 3D printer that can print high-viscosity UV-curable resins with a large exposure in the horizontal plane. The technical innovation is noteworthy, and the results could have implications for the design of new printing strategies to increase the throughput of 3D printing.

However, there are some areas that need improvement in the paper. For instance, more detailed information should be added to the experimental section, particularly regarding the resin preparation. Also, the English used in the paper needs to be improved.

Furthermore, there are some questions that the authors should address. For instance, is there any bubble inside the resin during the printing process? Also, for equation 2, is there any relationship between the detachment force and the detachment speed, which is crucial for the printing speed? The authors claimed the high throughput of 3D printing using LSVP, but they should compare the printing speed of LSVP and CLIP when printing the object with the same size. Additionally, there is no quantitative data for the serving life of the film.

Since the main innovation of the paper is the realization of 3D printing of resins with high viscosity, the authors should provide more characterization data for the printed objects using high viscous resin. The DMA data, the morphology of the curing part, the double bond conversion of the 3D printed objects, the printing accuracy, the curing depth and minimal exposure dose, volume shrinkage, et al, should be added.

Lastly, the authors should explain why they solely used ACMO with mono functionality and whether they tried to use diacrylate as a diluent as well as a crosslinker.

Overall, the paper presents a new and exciting development in the field of 3D printing, but some revisions and additions are necessary for a more comprehensive understanding of the results.

Response to Reviewers

We thank the reviewers for providing constructive feedback of our submitted work. We have revised the manuscript extensively and adequately addressed the reviewers' comments. In the following, the comments proposed by the reviewers are presented in blue italics for your convenience. After every comment is followed by our response.

We really appreciate the valuable suggestions proposed by the reviewers, which help us a lot in improving the scientific quality of our paper.

REVIEWER COMMENTS:

Reviewer #1 (Remarks to the Author):

The paper presents a line scan-based vat photopolymerization (LSVP), which can accomplish "spreading", "curing", and "detachment" simultaneously. This idea has been demonstrated for the same purpose and presented before in a paper, "Linear Immersed Sweeping Accumulation" in the Journal of Manufacturing Processes in 2016. The authors need to compare their approach with the one presented in the JMP paper. The scanning method used is also similar to EnvisonTec's vector 3SP technology. The authors need to articulate the novelty of the paper and how it is different from the previous work as well as why their approach is better.

Response: Thank you for the suggestions.

JMP published "LISA: Linear Immersed Sweeping Accumulation" [*Journal of Manufacturing Processes* 24 (2016): 406-415.] can continuously fabricate layers of 3D models without additional recoating time by **immersing the linear light source in liquid resin**. This novel AM process of LISA has advantages both in building time and flexibility.

Through the comparison, LSVP presented in this work is intrinsically different from LISA. It can be summarized from following aspects:

- From the aspect of fabrication fashion, LSVP adopts a "bottom-up" method, whereas LISA adopts a fabrication fashion similar to "top-down" approach, but the layer-by-layer accumulation is realized by the movement of the light source instead of the building platform.
- From the aspect of laser module, although LISA and LSVP both employ a "linear scanning" strategy, the gyro mirror systems of which are different. LISA employs a linear laser modulus with the combination of two-axis gyro-mirror system. LSVP employs a reflecting prism which has a constant rotation speed of 16,000 rpm. Compared to two-axis gyro-mirror system, reflecting prism is lower cost and more facile to use.
- From the aspect of applied materials, according to fabrication mechanism described in the paper, applied materials for LISA should be low viscous, since the refill of resin is highly dependent on the fluidity of the resin. Yet LSVP is capable of printing high viscosity resin by employing rollers assisted recoating mechanism.

The reviewer also mentioned vector 3SP technology proposed by EnvisionTEC. Similar to LISA, vector 3SP technology employs a top-down fashion. This technology is similar to stereolithography (SLA), that is the leveling of the resin is realized by a recoater. It has been generally considered that a resin with a viscosity exceeding 5 Pa s is not recommended printed through this approach [*Inst. Mech. Eng., Part C*, **2003**, 217(1), 105.][*Adv. Mater.*, **2020**, 32(25), 2001646.].

In conclusion, compared to LISA and vector 3SP, the features of LSVP technology not only limit in the linear scanning strategy, but also its way of refilling the resin on the film. Meantime, compared to the “top-down” fashion which was adopted by LISA and vector 3SP, the “bottom-up” fashion accompanied with a linear laser module used in LSVP system has following advantages:

- The capability of printing high viscosity UV-curable resin (600,000 cps presented in this work);
- The building size, especially in z axis, is higher;
- Only few initial resins are needed to realize the 3D printing, which is friendly for the development of high-performance polymer material. In our work, only 60-100 grams of resin is sufficient for printing samples including dog bone tensile test specimen and flexural specimen.

Corresponding statement had been compiled in the revised manuscript (*Section 2.1 “Prototype Assembly”*) and highlighted. Also, relative references have been cited in the revised manuscript.

Extracted from the revised manuscript:

... The linear scanning strategy, has been previous reported by Mao et al. used in a “LISA” system³², which can effectively eliminate the recoating and separation time. Also, such strategy was also employed in vector 3SP 3D printer developed by EnvisionTEC, which delivers a higher resolution and larger build envelope. However, reported linear scanning strategy is only suitable for the “top-down” fashion 3D printer, while the printing of high viscosity UV-curable resin has not been reported. ...

... to eliminate the uneven energy distribution errors brought by the start and stop time of conventional two-axis gyro-mirror system, in this work, a reflecting prism was...

Also, some more details of the method need to be presented.

(1) The dimensions in Fig. 1C need to be given. What are the sizes of the rollers, and how much are the rollers raised? This will give me an idea of how much the FEP film is deformed.

Response: Thank you for your valuable suggestions. The diameter of the rollers is 6 mm, and the linear shafts are theoretically designed to raise 2 mm higher than threaded rollers. The illustration presented in the manuscript is only for a better understanding for the readers. Hence the deformation is exaggerated. The engineering drawing is specified in the revised **Supplementary Information (Fig. S1)**. Also, the deformation of the FEP film in z axis is 9 mm.

For your convenience, the diameter of the thread rollers along with the linear shaft and the exact size of the raised rollers are given as below (unit: mm):

Fig. S1 Engineering drawing of four rollers

The illustration of installation of resin tank is shown as below (**Fig. R1**):

Fig. R1 Illustration of resin tank installation

(2) The tightening of the FEP film from all four sides is required to ensure no resin leaking. Under the film's tension, how much force is needed to deform the film in both X and Y directions is unclear. How to handle the contact points in the Y direction, which seems to have a significant discontinuity?

Response: Many thanks for the reviewer's professional comments. In the LSVP process, antileakage of the resins is one of our priority concerns. In the LSVP system, if the thread rollers are directly cast on the flat FEP film, the force to deform the film in both X and Y directions would be extremely high, exceeding the limitation of the FEP film. To handle this, we tentatively fold the flat FEP film into a box shape with a three-dimensional structure (refer to the figure below) when designing the resin tank. The detailed installation of the resin tank is illustrated in **Supplementary Movie S2**. This approach provides an effective solution from resin leakage, also reduce the tension force to the maximum extent.

Fig. R2 Detailed structure of the resin tank and installation demonstration

We also agree with the reviewer that the handling of the contact points is very important. To reduce the pressure, we cover a retaining washer on the tip of the threaded rollers as shown in the following figure (**Fig. R3**). We apologize for the ambiguous expression in our discussion and the schematic diagram, which makes the reviewer think that it is discontinuous in the Y direction, yet in fact the whole film is continuous. Corresponding revisions have been added in the manuscript. And we believe that the figures (**Fig. R2 & R3**) combined with the video

(Supplementary Movie S2) will present a clear explanation of the LSVP working mechanism.

Fig. R3 Contact part design of the threaded rollers.

(3) How such bending forces and continuous scratching/movement will affect the FEP film's life?

Response: Thank you for your question. As mentioned above, the flat FEP film was folded into a box shape with a three-dimensional structure to reduce the bending forces. Thereby the force and continuous scratching/movement exert less influence on the FEP film's life, and the fractures of the FEP film are mainly concentrated at the creases of the film. Nevertheless, it is encouraging to find that the FEP film was durable enough to support the printing of the Eiffel Tower (0.1 mm thick slice, 1020 layers in total) by both commercialized low viscosity and our high viscosity resin. The theoretically service-life of the FEP film is highly depends on the model, which will be further researched in our future improvement.

(4) The surface portion between the curing line and the left linear shaft in Fig 1C will attach to the cured layer. What are the dimensions? How big is the dragging force in the X direction?

Response: The distance between the two rollers is 20 mm. The length of the linear shaft is 130 mm. Yet the width of the platform is 100 mm and the maximum width in Y direction is 108 mm. Hence, the maximum value of dimensions is 10 mm × 108 mm. The dragging force in the X direction is closely related to the elastic modulus value after the resin is cured. Usually, higher elastic modulus brings a higher dragging force in the X direction.

At current stage, there is no well-established method to evaluate the dragging force precisely. As expounded in *Section "Analysis of LSVP system"*, a set of fixtures that can be conducted on a universal testing machine (Fig. 2) was designed to determine the pulling force, and to compare the release differences between LSVP and conventional DLP printers. Under the same curing area, the highest value of direct pulling force using conventional DLP 3D printer reaches 129 N, whereas the pulling force using LSVP 3D printing is ca. 16 N, indicating a lowered force in LSVP system. Corresponding physical photos are shown in Supplementary Information (Fig. S5).

Fig. S5. Physical photos of detachment force tests setups. (A) For conventional DLP 3D printer simulation. (B) For LSVP system simulation.

(5) The detaching force measurement in Fig. 3C differs from the setup shown in Fig. 1.

Response: In view of that the detaching force cannot be determined through *in-situ* characterization so far, a set of fixtures (Fig. 2C (we remove original Fig. 2 to the supplementary information, which makes the original Fig. 3C changed to Fig. 2C in the revised manuscript)) suitable for the universal testing machine to measure the detachment force was designed. Although this is somewhat different from the practical test situation, it provides evidence of the differences in detaching force between LSVP and DLP 3D printing technology. Enlarged diagram were captured in the following figures:

For DLP 3D printer:

Fig. R4 Clamp design for DLP 3D printer

For LSVP 3D printer:

Fig. R5 Clamp design for LSVP 3D printer

By comparison, we found that the release force of LSVP is much lower than that of conventional DLP molding methods on large planes, which proves the advantages of LSVP.

Reviewer #2 (Remarks to the Author):

High-viscosity resin 3D printing is of interest but challenging. This paper develops a VPP method for viscous resin printing by employing a line-shape laser system and four rollers to create an isolated print area that eases the detachment of part from substrate. The claim for being able to print resin with a viscosity of > 600,000 cps is impressive compared to traditional limit of 5,000 cps. However, the introduction section is poorly written without accurate introduction about the fundamentals, without a comprehensive review and comparison to current state of work, and without a coherent flow of information and motivation. Further, the results and discussion section miss details on some key mechanisms.

Response: We would like to thank the reviewers for pointing out our writing deficiencies in the introduction part. In the revised manuscript, we reorganized the introduction section to highlight the motivation and novelty of our presented work. Revised portions are highlighted in the new manuscript.

Please check the comments as follows.

1. Lines 44-51 introduce the DLP's advantage and limitation. Some descriptions or comparisons to SLA are inaccurate. E.g., the paper says DLP has limited throughput, but it is known that DLP is faster than SLA. The authors should also note that not only DLP but also SLA require repeated movement of the build platform. This paragraph needs to be modified to present a more accurate and fair comparison of DLP with other VPP methods and also a comparison of VPP methods with other AM methods.

Response: We thank the reviewer for correcting our inaccurate description in the literature review section. DLP has limited throughput is that the platform has to ascend and descend upon every layer fabrication, which is time-consuming. We have made corresponding revisions in the introduction section. We agree with the reviewer that the DLP is faster than SLA, especially when CLIP technology was invented.

In various AM methods, VPP delivers the largest build envelop along with highest dimensional accuracy. VPP is usually categorized as bottom-up (represented by DLP) and top-down (represented by SLA). DLP is usually employed to fabricate something delicate with a higher printing speed (dental application for instance), while SLA is usually employed to build multiple larger outer shell parts in one batch.

Here are some contents extracted from the revised manuscript:

...Among various 3D printing technique, vat photopolymerization (VPP) is one of the most promising technologies² in view of its advantageous of high-efficiency, large build envelope, economic, and versatile material options³⁻⁷...

...Moreover, compared to SLA which employs a single laser point, DLP 3D printing technology achieves a resolution of about 1 μm regardless of the layer's lateral area and complexity¹⁰. Yet, there are still some challenges for DLP 3D printing technology waiting to be addressed. For example, both SLA and DLP 3D printing technologies are facing the limitation of printing time. ...

Btw, the literature review doesn't elaborate on the importance or the need for printing high-viscosity materials. E.g., what are the potential applications given an ability to print viscous materials? The authors should convey the motivation of printing viscous resin better.

Response: We also thank the reviewer for pointing out the deficiencies in the motivation presentation of this work. This proposed project is based on the development of 3D printed elastomers by VPP, which can be applied in the flexible sensors, soft robots preparation and so on. It has been uncovered that printing oligomer dominated high-viscosity UV-curable resin is the most effective and direct way to improve the mechanical properties of the resulting elastomers [*Advanced Materials* 2017 Vol. 29 Issue 15 Pages 1606000]. Although UV-curable resins with superior mechanical properties (e.g. Ebecryl® 8413) have been developed, those viscous resins are heavily diluted for practical applications in 3D printing, resulting in a mechanical properties deterioration.

The reason of the deterioration is the insufficient molecular weight growth. Limited by the free radical polymerization mechanism of the UV-curable resin material used in VPP, the preparation of elastomeric materials is still a bottleneck in the current VPP manufacturing technology because it cannot achieve rapid growth in molecular weight upon UV exposure. Prof. Timothy E. Long in Virginia Tech University pointed out in his review [*Progress in Polymer Science* 2019 Vol. 97 Pages 101144] that the complication with VPP involves the fundamentals of rubber elasticity and its implications on viscosity. Although using a recoater to assist the leveling of high-viscosity resins can increase the operable viscosity of the original UV-curable resin from 5 Pa s to 18 Pa s, the process of using a scraper to assist in applying the resin is still full of challenges. Because the low storage modulus of the cured network cannot withstand the shear force during recoating, resulting in deformation or even collapse of the formed structure. The core problem of printing high-viscosity UV-curable resins is the difficulty in the leveling of the resin. If the high-viscosity resin containing Newtonian and non-Newtonian fluids can be leveled in the uncured area efficiently and uniformly in a short time, it could be able to prepare elastomers with superior mechanical properties by VPP.

Hence, we were trying to break through the bottleneck problem of conventional leveling method, and use the large deformation method to achieve the leveling of ultra-high viscosity resin, so the prototype of LSVP technology is gradually formed. This developed LSVP system can be extended to prepare various materials with outstanding properties developed with 3D printing.

Corresponding revision relating to the motivation part had been added in the manuscript and highlighted. Thank you.

Here is part of the revised contents extracted from the manuscript:

... For example, the oligomers with high molecular weight facilitate the formation of highly entangled polymer chain, contributing to the preparation of high-elastomeric material with improved strain and stress. ...

2. Line 52: the authors might mention Carbon's technique of CLIP as well, which is well known for its breakthrough in VPP.

Response: Thank you for your suggestion. We agree with the reviewer that the CLIP technique invented by Carbon 3D is an important breakthrough in VPP, which eliminates the ascend and

descend procedure during processing, thus greatly shortens the fabrication time [Tumbleston, John R., et al. "Continuous liquid interface production of 3D objects." *Science* 347.6228 (2015): 1349-1352.]. The revision has been made in the revised manuscript.

3. Please check and fix all the grammar errors.

E.g., Line 64: "odor" is mistakenly used as an adjective.

Line 68: In "yet the extremely high viscosity of which limit their printability in currently used 3D printer", should "limit" be limits and "printer" be printers?

Lines 69-71: it is hard to understand this sentence: "By eliminating the oxygen inhibition effects, bottom-up fashion provides higher selectivity for the materials, including acrylate monomer, epoxy acrylate oligomer and polyurethane acrylate oligomer and so on."

Line 78: "the DLP printing elastomer"

More other errors throughout the manuscript. The authors should thoroughly proof it.

Response: Thank you for pointing out our grammar errors. We've polished the manuscript thoroughly. Those corrections were highlighted in the revised manuscript.

Here are some representative revisions extracted from the revised manuscript:

- ...Despite that, diluents and reactive monomer is volatile and irritating...
- ...yet the extremely high viscosity of which limits their printability in currently used 3D printers.
- Currently, the UV-curable acrylate resins for bottom-up fashion are roughly categorized as monomer, epoxy acrylate oligomer and polyurethane acrylate oligomer...
- Although the elastomers prepared by VPP technique has been reported, ...

4. Line 102 (also Line 52) about the paper's goal: What is the specific definition or metric of "throughput" in the authors' goal here? Print speed? Print quality (if so, name the metrics)? If the work aims to increase the print speed for high-viscosity resin, what is the print speed for viscous resins reported in current literature? How does this LSVP speed compare to current speed? Or if throughput means other properties, provide a clear comparison.

Response: In this manuscript, the "throughput" represents the "print speed".

The goal of this work is to print high viscosity resin as mentioned in the title, and the designed LSVP system aims to strike a balance between effective and efficiency, not only to process ultra-high viscosity UV-curable resin, but also ensuring that the efficiency is maintained. However, for high-viscosity printing, especially the printing of high-viscosity Newtonian fluid, we have not found any relevant literature reports at present, because the resin with a viscosity as high as 600,000 cps basically cannot be printed directly. Usually, the printing of these high viscosity resins is realized by DIW technology, which also require a fast viscosity response when shear force was applied on the syringe. Meantime, DIW technology is incapable of printing complex structured parts. In LSVP, we've carried out corresponding experiments and it took about 7 hours to print an Eiffel Tower using such high-viscosity resin (Fig. 4a).

To make a print speed comparison, the Eiffel Tower was further printed with low viscous resin (ca. 600 cps) by LSVP, which took about 5 hours (corresponding video clip has been

uploaded as **Supplementary Movie 6**). Snapshots are shown as below.

Fig. R6 Snapshots captured from Supplementary Movie 6

The print efficiency of LSVP is superior to that of LCD 3D printing technology (similar to conventional DLP technology), which takes 13 hours to print Eiffel Tower with similar size (<https://www.youtube.com/watch?v=snuwqJU6nvs>), but is inferior to that of the CLIP technique (6 and a half minutes to print Eiffel Tower with similar size, <https://www.youtube.com/watch?v=KQIU7AluWMo>).

The advantage of LSVP is not in efficiency of printing low-viscous resins compared to CLIP technology, while in processing high-viscous resins, LSVP exhibits a relatively high printing speed, since there is no significant difference in the time for LSVP to print high-viscos resins (7 hours) and low-viscos resins (5 hours). Therefore, we believe that the LSVP is a promising technique for efficient processing of ultra-high viscosity UV-curable resins.

5. *Addressing the following issues is crucial to accomplishing the LSVP.*

Is the light intensity profile uniform as the laser line moves across the build platform? Any projection gap between the scanning laser lines? The authors should discuss the exposure profile in detail.

Response: Yes, the light intensity profile is uniform as the laser line moves across the build platform. In LSVP system, the only variable parameter is “number of scans” (as shown in the **Fig. R7** & **Fig. S8**). The minimum number of scans in LSVP system is set as 5, which can avoid the projection gap between the scanning laser lines.

Fig. R7 Illustration for model slicing strategy

Fig. S8 Illustration of scanning strategy

Also, the synchronization between the roller and laser projection needs to be elaborated. How to determine an optimal laser line exposure time and scan speed?

Response: The roller and laser modulus are fixed together physically and as shown in the followed pictures. This mechanical approach ensures the synchronization of the roller and laser projection without the need for additional calibration.

Fig. R8 Illustration of fixing of linear laser modulus

As for the determination of optimal laser line exposure time and scan speed, we apologize for the ambiguity of the expression. In LSVP system, the “scan speed” representing the movement speed of laser module in x direction (shown in the Fig. 1c. in the revised manuscript.) is only related to the “number of scans”. In this prototype design of LSVP, if the number of scan is 1, the movement of the laser modulus is 35 mm/s. While increasing the “number of scans” to n , the movement speed of the laser modulus $v = \frac{35}{n} \text{ mm} \cdot \text{s}^{-1}$. As shown in the figure that, an increased “number of scans” represents a stronger UV-exposure dose. Yet, increased “number of scans” also indicates a slower movement of the laser module, which results in a longer printing time.

Corresponding illustration has been added in the revised **Supplementary Information**.

For the optimization of “number of scans”, here a “BJA-27.stl” printing file was employed to evaluate the proper print parameters. This .stl file had been uploaded in the submission system. For different types of UV-curable resins with different reactivity, the “number of scans” should be adjusted individually. If the UV exposure is proper, the edge of each pillar should be distinctive, whereas excessive scanning resulting in a connection between the pillars.

Fig. R9 Illustration of precision test

Will the rolling laser module jitter due to the flow resistance from the very viscous resin? Any vibration or non-steady movement of laser module during LSVP? If so, motion blur could be induced to the projected laser line and cause loss of print resolution. Besides, the possible jittering of laser scans may affect the print surface roughness and geometrical features.

Response: As mentioned above, the rolling laser module is fixed on a lead screw, ball-screw more precisely. This design will eliminate jitter nor non-steady movement of the laser module during LSVP. As shown in **Fig. R10**, LSVP is capable of printing refined lattice structure.

Fig. R10 Printing of refined lattice structure by LSVP.

6. *Some content in Section 2 seems to fit in Section 4 instead.*

Response: Thank you for your valuable suggestion. We've rearranged the sample preparation description part in *Section 2* to *Section 4* (Methods) for a better understanding.

Here are the extracted parts which was moved to *Section 4* (Methods):

... For the preparation of the homogeneous high viscosity resin, TPO was first dissolved in ACMO by stirring for 2h in a dark room, then mixed...

... After printing, the samples fabricated from the low viscosity resin, isopropanol was sufficient to wash away the remaining uncured resin. ...

7. *What is the print speed and resolution of LSVP?*

Response: As mentioned in the previous question, the printing speed is closely related to the model. Taking the ordinary photosensitive resin model mentioned in the text as an example, the printing time of an Eiffel Tower is 5 hours. The accuracy of LSVP on the Z axis is closely related to the accuracy of the Z axis moving screw. We have achieved the control of the layer thickness of 0.025mm on the Z axis stepping motor, which can be seen in the image captured by scanning electron microscope. The resolution in the XY direction depends on the spot size of the laser, and the current spot is controlled at about 0.15mm.

8. *Any recommendation for future work to improve the LSVP?*

Response: Thank you for your question about the future improvement of LSVP. In future, we will give full play to the advantages of LSVP, including:

- **Building Size.** Compared with DLP, the advantage of LSVP is that it does not need a large depth when printing large format. The width of build envelope increased from the original 80 mm to 280 mm. When developing the large-format LSVP system, we placed the linear laser horizontally, and changed the vertical direction of the laser through a 45° angle. It can be seen from the figure that such an LSVP system does not need to be grounded to achieve short-focus printing. The above work is in the process of sorting out and optimizing. This is the current direction for us to improve the LSVP system in terms of equipment and materials.

Fig. R10 Enlarged setup for LSVP system with a smaller footprint.

- 3D Printed Elastomer Preparation through High Viscosity Resin. 3D print high-performance UV-curable resin is the goal for the development of LSVP. At this stage, by synthesizing blocked amine polyurethane acrylate, we have prepared an elastomer material with an elongation at break of more than 1600% and a tensile strength higher than 25 MPa based on dual-curing mechanism. These data come from samples which is directly 3D printed, not samples prepared by casting method. At the same time, the material has excellent resilience performance. After repeated compression and rebound, the deformation rate of the material is less than 5%, which is even better than the performance of some polyurea elastomers. This work is still on-going.

[REDACTED]

Reviewer #3 (Remarks to the Author):

Wu et al have presented an interesting academic paper describing a new 3D printer that can print high-viscosity UV-curable resins with a large exposure in the horizontal plane. The technical innovation is noteworthy, and the results could have implications for the design of new printing strategies to increase the throughput of 3D printing.

However, there are some areas that need improvement in the paper. For instance, more detailed information should be added to the experimental section, particularly regarding the resin preparation. Also, the English used in the paper needs to be improved.

Response: Thank you for your suggestions. More detailed information relating to resin preparation has been added in the revised manuscript. Also, we have polished the manuscript thoroughly.

Furthermore, there are some questions that the authors should address. For instance, is there any bubble inside the resin during the printing process? Also, for equation 2, is there any relationship between the detachment force and the detachment speed, which is crucial for the printing speed?

The authors claimed the high throughput of 3D printing using LSVP, but they should compare the printing speed of LSVP and CLIP when printing the object with the same size. Additionally, there is no quantitative data for the serving life of the film.

Response: The preparation of the high viscosity resin is conducted on a planetary centrifugal mixer, which is a machine designed for mixing paste uniformly. Also, this planetary centrifugal mixer is equipped with a vacuum pump which can further assist to remove the bubbles in the resin system during the mixing. Corresponding preparation procedure has been added in the revised manuscript. The bubbles may be introduced during the pouring of the resin in the resin tank, that's why we adopted threaded rollers where the grooves on the threaded rollers surfaces which attached on the upper surface of the film can remove the bubbles effectively (see attached figure in the left). From the SEM image (see attached figure in the right) of the elastomer printed by high viscosity resin, no obvious bubbles can be observed. Please see **Supplementary Movie 3** for more information.

Fig. R12 Carved grooves on the threaded rollers and SEM images for printed lattice structure.

For Equation 2, we refer the model mentioned in the literature [*Journal of Physics D: Applied Physics* **1971** Vol. 4 Issue 8 Pages 1186].

[REDACTED]

Fig. R13 The peel test captured from [Fig. 8. from *Journal of Physics D: Applied Physics* 1971 Vol. 4 Issue 8 Pages 1186.]

The cited paper also concluded the equation which is

$$P = \frac{b\gamma}{1 - \sin \theta} \quad (23)$$

According to this equation, no variate representing the detachment is shown. Although the reference mentioned that “as expected the force required to peel the film from the substrate depended markedly on the peeling velocity.”, “The dependence of peel force on peeling velocity was attributed to viscous processes related to the rate of separation of interfacial bonds.” In this work, after the curing of the resin, the film and the cured resin part can be treated as a rigid plastic, which is not suitable to the “viscous processes”. Due to that, we think that at low detachment speed the detachment force tended to a constant value according to the equation. We want to emphasize that the adoption of these two equations is to investigate the differences of detachment force between DLP and LSVP 3D printers. Noted, the detachment speed for both experiments are set to 2 mm/s.

As for the high throughput, the print speed of LSVP is higher than conventional DLP/LCD 3D printer (<https://www.youtube.com/watch?v=snuwqJU6nvs>), however, not comparable to CLIP technology when low-viscous resins were applied (<https://www.youtube.com/watch?v=KQIU7AluWMo>). For example, the printing of an Eiffel Tower, according to the online video, it only takes 6 mins. However, in LSVP system, it will take hours (Please see added **Supplementary Movie 6**). However, the printing speed of CLIP technology is highly dependent on the exposure area. That is to say, if a solid part (like shown in **Fig. 4b**) was printed by CLIP technology, it will take longer, because it needs longer time to flow the resin into the tiny gap, which makes viscosity of the resin lower than 500 cps [*Biomacromolecules*, 2019. 20(4): p. 1699-1708.]. However, the efficiency of LSVP system will not be affected by the viscosity, which is only affected by the reactivity of the used resin. In processing high-viscosity resins, LSVP exhibits a relatively higher printing speed since there is no significant difference in the time for LSVP to print high-viscos resins (7 hours) and low-viscos resins (5 hours). Therefore, we believe that the LSVP is a promising technique for efficient processing of ultra-high viscosity UV-curable resins.

As for the quantitative data for the serving life of the film, in this test, the film can at least print a large lattice structure (**Fig. 6c**) and Eiffel Tower (**Fig. 6d**). Actually, the service life of the film is highly relied in the printed model. More importantly, the service life of the film depends on the hydrophobicity of the film. Recently, we applied modified POSS on a FEP film to obtain a highly transparent and omniphobic film, which showed a very low contact angles hysteresis

(CAH) for resins, which means the tiny tilt angle will drive the flow of the high viscosity resin, and smaller force is needed to realize the detachment of the cured resin from the film. In the future work, we'll apply this film on the LSVP system to evaluate the improvement of the service life of the modified film.

Fig. R14 Pre-experiment for CAH test of the omniphobic film and UV-vis spectrum of the film

Since the main innovation of the paper is the realization of 3D printing of resins with high viscosity, the authors should provide more characterization data for the printed objects using high viscous resin. The DMA data, the morphology of the curing part, the double bond conversion of the 3D printed objects, the printing accuracy, the curing depth and minimal exposure dose, volume shrinkage, et al, should be added.

Response: Thank you for your valuable suggestions. In the revised version, we have added DMA data, double bonds conversion rate, curing depth evaluation and other data to further characterize the property of oligomer dominated UV-curable resin. These data are compiled in the supplementary information (**Fig. S8-S15**) and corresponding explanation are added in the revised manuscript.

In brief, both CN8888NS and CN8883NS series samples exhibit similar trend, that is, higher proportion of oligomer leads to a lower volumetric shrinkage. It also should be noted that although increase of oligomer content will decrease UV-curing polymerization rate, from the FTIR spectrum, high conversion rate of C=C double bonds was achieved by LSVP system according to the absence of C=C characteristic peaks upon UV exposure. Meantime, from the DMA results, higher proportion of monomer brings a higher T_g , which deteriorates the resilience of the elastomer.

Here are some representative data including FTIR, volumetric shrinkage and DMA curves of CN8888NS series samples:

Fig. R15 Additional characterization of CN8888NS series samples

Lastly, the authors should explain why they solely used ACMO with mono functionality and whether they tried to use diacrylate as a diluent as well as a crosslinker.

Response: The implementation of this project is to prepare high-performance elastomer through VPP technology. High-performance VPP printed elastomers are highly rely on the molecular weight, and the resultant increased viscosity is way exceed the conventional 3D printer limitation[*Progress in Polymer Science* **2019** Vol. 97 Pages 101144]. LSVP was initially expected to solve the difficulties of printing high viscosity resin. It had been reported from 2017 [*Advanced Materials* **2017** Vol. 29 Issue 15 Pages 1606000] that in order to prepare elastomers by VPP, the combination of diacrylate oligomers and monomers is an effective way to balance the strain of printed elastomer. Similar monomers including hydroxypropyl acrylate (HPA) [*ACS Applied Materials & Interfaces* **2022** Vol. 14 Issue 9 Pages 11727-11738], 2-hydroxyethyl acrylate [*ACS Macro Letters* **2019** Vol. 8 Issue 11 Pages 1511-1516] and other monomers for 3D printing elastomer have been reported. In our previous study, diacrylate reactive diluents including HDDA, PEG200DA, DPGDA were introduced to the resin system together to balance the viscosity. But those reactive diluents severely affect the strain of the printed elastomer. Therefore, diacrylate was not adopted in this work. Also, we've studied the effects brought by various monomers on the mechanical properties of 3D printed elastomers. These monomers include CTFA, IBOA, ACMO, EHA, EHMA. In our study, we found that EHA and EHMA can provide a favorable resilience. However, the tensile strength and elongation at break are not impressive. Although IBOA along with IBOMA provide comparable physicochemical mechanical properties, the odor makes them out of our considerations. Polymerized ACMO is a hydrophilic polymer and is known to be non-toxic, non-antigenic and biocompatible. [W. Li, M. Nakayama, J. Akimoto and T. Okano, *Polymer*, **2011**, 52, 3783–3790][H. Takahashi, M. Nakayama, K. Itoga, M. Yamato and T. Okano, *Biomacromolecules*, **2011**, 12, 1414–1418] ACMO is not only an excellent diluent for oligomer, but also conducive to maintaining the mechanical properties (higher elastic modulus) of the printed elastomer. Considering above factors, in this work, ACMO was adopted to diluent high viscosity resin.

Corresponding explanations are added in the revised manuscript.

Overall, the paper presents a new and exciting development in the field of 3D printing, but some revisions and additions are necessary for a more comprehensive understanding of the results.

Thank you for your positive comments on our work, we have revised the manuscript thoroughly for a better understanding. Also, we will keep on optimizing the LSVP system to fabricate 3D printed parts with superior mechanical properties.

REVIEWERS' COMMENTS

Reviewer #1 (Remarks to the Author):

The paper presents a line scan-based vat photopolymerization (LSVP), which can accomplish "spreading", "curing", and "detachment" simultaneously. This principle is similar to "Linear Immersed Sweeping Accumulation" in JMP in 2016 and has been demonstrated before.

A different scanning method is used based on a reflecting prism; however, this scanning method is exactly the same as EnvisonTec's vector 3SP (Scan, Spin, and Selectively Photocure) technology (refer to <https://www.youtube.com/watch?v=XQXWiR1b0ek>), which is commercialized about 7-8 years ago and has been well known in the field (refer to: <https://asmedigitalcollection.asme.org/manufacturingscience/article/136/6/061023/377632/A-Direct-Tool-Path-Planning-Algorithm-for-Line> for more insight on 3SP). EnvisonTec has developed both top-down and bottom-up versions using this light engine.

Finally, the bottom-up printing using a moving laser and the rollers-assisted recoating mechanism used in the paper is very similar to Formlab's Form3 printer, which was commercialized about 4 years ago (refer to: <https://www.youtube.com/watch?v=D6Gbn2zOVL4>). The Form3 printer is now widely available in universities. The paper claims high-viscosity resin printing. But from the printing method perspective, it is unclear what makes it differ from Form3 (or why Form3 cannot print resins with similar viscosity?).

Overall, I applaud the authors' efforts in revising the paper and think the presented FEP-based roller design has some technical merits; however, the novelty of the paper is limited. Its impact on the research community is small, not at the level of Nature Communications.

Reviewer #2 (Remarks to the Author):

This revision is improved based on the response and updates. No further comments.

Reviewer #3 (Remarks to the Author):

The revised manuscript has fully addressed the concerning points and can be accepted in its current form. Congratulations on this accomplishment.

Response to Reviewers

Reviewer #1 (Remarks to the Author):

The paper presents a line scan-based vat photopolymerization (LSVP), which can accomplish "spreading", "curing", and "detachment" simultaneously. This principle is similar to "Linear Immersed Sweeping Accumulation" in JMP in 2016 and has been demonstrated before.

A different scanning method is used based on a reflecting prism; however, this scanning method is exactly the same as EnvisionTec's vector 3SP (Scan, Spin, and Selectively Photocure) technology (refer to <https://www.youtube.com/watch?v=XQXWiR1b0ek>), which is commercialized about 7-8 years ago and has been well known in the field (refer to: <https://asmedigitalcollection.asme.org/manufacturingscience/article/136/6/061023/377632/A-Direct-Tool-Path-Planning-Algorithm-for-Line> for more insight on 3SP). EnvisionTec has developed both top-down and bottom-up versions using this light engine.

Finally, the bottom-up printing using a moving laser and the rollers-assisted recoating mechanism used in the paper is very similar to Formlab's Form3 printer, which was commercialized about 4 years ago (refer to: <https://www.youtube.com/watch?v=D6Gbn2zOVL4>). The Form3 printer is now widely available in universities. The paper claims high-viscosity resin printing. But from the printing method perspective, it is unclear what makes it differ from Form3 (or why Form3 cannot print resins with similar viscosity?).

Overall, I applaud the authors' efforts in revising the paper and think the presented FEP-based roller design has some technical merits; however, the novelty of the paper is limited. Its impact on the research community is small, not at the level of Nature Communications.

Response:

Many thanks to the reviewer's professional comments. The works mentioned above are important breakthroughs in the field of vat photopolymerization. We have checked the link reviewer presented carefully and respectfully clarify the distinguish of our work below.

1. Compared with the reported LISA in JMP in 2016, both LISA and LSVP adopts linear scanning strategy to realize 3D printing. However, LISA adopts two gyro mirrors, while LSVP adopts single mirror which is more facile and stable. LISA adopts immersive printing approach which is ascribed to the top-down strategy, while LSVP adopts a bottom-up approach.

2. Compared with 3SP technology, including types of Vector, Xtreme and Xede, (<http://www.abrasiveinnovations.biz/news/ultimate-guide-to-3d-printing-thermosets>), LSVP adopts similar linear scanning strategy. However, with the combination of large film deformation, LSVP is capable of printing high-viscosity resin. This is the merits of LSVP system.

3. The reviewer also mentioned the similarities between the Form 3 system and this work. The Low Force Stereolithography (LFS technology) adopted by the Form 3 system drastically reduces peel forces for significant increases in print quality and printer reliability. It should be noted that as stated on the Form 3 website (<https://formlabs.com/blog/form-3-form-2-3d-printer-comparison/>), the key to the LFS technology lies in the development of low release force which puts forward higher requirements for the low surface energy modification of the film while ensuring its UV transmittance. The deformation of the film is realized by the bump caused by the laser modulus. This release method puts forward higher requirements on the tension and flatness

of the membrane, that is, the mechanical properties of the membrane. We believe that Form 3 can realize the high-speed printing of resins with high modulus upon curing. Yet for the elastomeric parts with low green modulus, when the film does not have sufficient recovery force during long-term use without external deformation, it is still difficult to achieve effective separation. Take Flexible 80A released by FormLabs as an example, as described on its website: "Flexible 80A Resin is a high-viscosity resin. It may take longer to fill the resin tank than other resins. Tall, thin features are difficult to print in Flexible 80A Resin because the material flexes during the peel process. If possible, orient the part closer to the build platform but at a 20° angle minimum." (https://support.formlabs.com/s/article/Using-Flexible-Resin?language=en_US). In our experiments, LSVP system can realize the printing of large solid parts (no need for angle placement design) and fine structures (500 microns) rod structures for high-viscosity and low-modulus UV-curable resins. It also has been proved by printing Eiffel Tower (10 cm height) with fine structure that the LSVP system can print high-viscosity UV-curable resin by striking a balance between height and accuracy.

Overall, we want to point out that the focus of this work is to introduce an approach for 3D printing of high-viscosity resins by integrating an economical one-dimensional scanning laser system and a large deformation of the release film. The merits for printing high-viscosity is to realize a high extent of molecular weight growth, which is realized by LSVP system without complicated mechanical structure design.

Specifically, the proposal of this project is based on exploring how to print high-strength and high-elastic materials by vat photopolymerization system. The performance of the elastomer material heavily depends on the performance of the oligomer, while the high-performance oligomer, especially the dual-stage curing acrylate oligomer terminated with tert-butyl hindered amine, exhibits a very high viscosity at room temperature. The weight proportion of this oligomer in the resin composition determines the performance of the printed objects. We found that when the content of the weight ratio of the oligomer reaches higher than 70 wt%, it has far exceeded the upper limit of the acceptable viscosity of the current 3D printing equipment, yet the LSVP system can effectively address this printing difficulties.

It has to be admitted that LSVP also has its shortcomings. Although the linear laser has the advantages of low cost and long life, it has not undergone effective optical shaping, and its spot diameter cannot be effectively controlled below 0.2 mm which can't hold a candle to high-resolution 4K projector. Secondly, the raw material supplement approach of LSVP has not been mentioned in this work. We believe the above problems can be solved by modifying the instrument. What we want to emphasize is that the proposed LSVP technology is to introduce a new idea and method for printing high-viscosity photosensitive resin. At the same time, its design idea is simple and easy to be implemented in the laboratory. No specific film was needed, and no complex laser system was required. We hope that the proposed ideas in this work can provide 3D devices with excellent performance for the research group that develops high-performance prepolymers without being affected by the performance of reactive diluents.

Reviewer #2 (Remarks to the Author):

This revision is improved based on the response and updates. No further comments.

Reviewer #3 (Remarks to the Author):

The revised manuscript has fully addressed the concerning points and can be accepted in its current form. Congratulations on this accomplishment.